

# Analysis of core genes for colorectal cancer prognosis based on immune and stromal scores

Yi Zhu, Yuan Zhou, HongGang Jiang, ZhiHeng Chen and BoHao Lu

Department of Gastrointestinal Surgery, The Affiliated Hospital of Jiaxing University, Jiaxing, China

## ABSTRACT

**Background**. Colorectal cancer (CRC) is one of the most common malignancies.An early diagnosis and an accurate prognosis are major focuses of CRC research. Tumor microenvironment cells and the extent of infiltrating immune and stromal cells contribute significantly to the tumor prognosis.

**Methods**. Immune and stromal scores were calculated based on the ESTIMATE algorithm using the sample expression profile of the The Cancer Genome Atlas (TCGA) database. GSE102479 was used as the validation database. Differentially expressed genes whose expression was significantly associated with the prognosis of CRC patients were identified based on the immune matrix score. Survival analysis was conducted on the union of the differentially expressed genes. A protein–protein interaction (PPI) network was constructed using the STRING database to identify the closely connected modules. To conduct functional enrichment analysis of the relevant genes, GO and KEGG pathway analyses were performed with Cluster Profiler. Pivot analysis of the ncRNAs and TFs was performed by using the RAID2.0 database and TRRUST v2 database. TF-mRNA regulatory relationships were analyzed in the TRRUST V2 database. Hubgene targeting relationships were screened in the TargetScan, miRTarBase and miRDB databases. The SNV data of the hub genes were analyzed by using the R maftools package. A ROC curve was drawn based on the TCGA database. The proportion of immune cells was estimated using CIBERSORT and the LM22 feature matrix.

**Results**. The results showed that the matrix score was significantly correlated with colorectal cancer stage T. A total of 789 differentially expressed genes and 121 survival-related prognostic genes were identified. The PPI network showed that 22 core genes were related to the CRC prognosis. Furthermore, four ncRNAs that regulated the core prognosis genes, 11 TFs with regulatory effects on the core prognosis genes, and two drugs, quercetin and pseudoephedrine, that have regulatory effects on colorectal cancer were also identified.

**Conclusions**. We obtained a list of tumor microenvironment-related genes for CRC patients. These genes could be useful for determining the prognosis of CRC patients. To confirm the function of these genes, additional experiments are necessary.

Corresponding author
Yi Zhu, zhuanwen456@163.com

## INTRODUCTION

Colorectal cancer (CRC) is a common malignant tumor (*Long, Lundsmith & Hamilton, 2017*). Despite advances in understanding the molecular mechanisms of CRC, it still has a high mortality rate. Guided treatment based on outcome prediction is an effective method to reduce the mortality of CRC (*Yu et al., 2017*). In recent years, studies on the prognostication of CRC have become increasingly important, and a large number of potential outcome predictors have been identified (*Sanz-Garcia et al., 2017*; *Rena et al., 2002*). The pathogenesis of CRC is comprehensive. Tumor cell intrinsic genes, especially master transcription factors, dictate the initiation, progression, and evolution of CRC (*Wang et al., 2017*; *Suvà et al., 2014*). The tumor microenvironment has also been reported to critically influence the expression of genes in tumor tissues and the prognosis (*Curry et al., 2014*; *Cooper et al., 2012*).

The tumor microenvironment is the cellular environment of tumor cells, consisting of extracellular matrix, soluble molecules and tumor stromal cells. In the tumor microenvironment, immune cells and stromal cells are the two main types of nontumor components and have been proposed to be valuable for the diagnosis and prognosis evaluation of tumors (*Fukumura et al., 2010*). In 2011, scientists began to examine various stages (TNMI-IV) of CRC for immunity to authenticate grading and staging systems, and there was considerable research on immune scoring stages and TNM staging. The results showed that an immune scoring system could predict the outcomes of CRC more accurately. It has obvious advantages for predicting survival in patients with CRC (*Mlecnik et al., 2011*; *Fridman et al., 2012*; *Galon, 2006*). According to the study conducted by *Yoshihara et al. (2013)*, immune and stromal scores could be used to predict the infiltration of nontumor cells by analyzing specific gene expression data from The Cancer Genome Atlas (TCGA) database of immune and stromal cells.

However, no research has been conducted on predicting the prognosis-related genes of CRC by using the tumor microenvironment score. For the first time, in this current work, by taking advantage of both the TCGA database of CRC cohorts and the ESTIMATE algorithm-derived immune scores, we extracted a list of microenvironment-associated genes that predict the outcomes of CRC patients.

## MATERIALS & METHODS

### Materials

(1) Gene expression profile data and the clinical information of 362 CRC patients from TCGA were obtained from the NCI Genomic Data Commons (https://portal.gdc.cancer.gov) (*Heath et al., 2021*), and samples with missing survival information or a follow-up of fewer than 30 days were excluded (Table 1).

(2) Gene expression profile data and clinical information from the GSE102479 dataset (152 CRC patients) were obtained from the Gene Expression Omnibus (GEO) database, and samples with missing survival information or a follow-up of fewer than 30 days were removed from the subsequent verification (Table 2).

**Table 1  Clinical Information of patients with colorectal cancer from TCGA.**

| Parameter | subtype | patients |
|---|---|---|
| Age | >66 | 175 |
| | ≤66 | 187 |
| Gender | Female | 160 |
| | Male | 202 |
| Stage | I | 55 |
| | II | 130 |
| | III | 110 |
| | IV | 50 |
| | Unknow | 17 |
| OS time(days) | >700 | 181 |
| | ≤700 | 181 |

**Notes.**
Abbreviations: OS, overall survival; TCGA, The Cancer Genome Atlas.

**Table 2  Clinical Information of patients with colorectal cancer from GEO.**

| Parameter | Subtype | Patients |
|---|---|---|
| Age | >70.1 | 75 |
| | ≤70.1 | 76 |
| | NA | 1 |
| Gender | Female | 68 |
| | Male | 84 |
| Stage | II | 80 |
| | III | 72 |
| OS time (months) | >51.2385 | 76 |
| | ≤51.2385 | 76 |

**Notes.**
Abbreviations: OS, overall survival; GEO, Gene expression omnibus.

## Methods

### Calculation of the immune matrix score and identification of differentially expressed genes (DEGs)

The ESTIMATE algorithm was applied to calculate the immune score and stromal score of the sample expression profile of the TCGA CRC patients. Maxstat software was used to find the best cut-off of the immune score and matrix score. Samples were classified into two groups with high and low scores based on the score of the best cut-off to analyze and calculate the DEGs. Limma was used to analyze the gene expression data after processing, and Log(Fold Change) > 0.2 and $P$ value < 0.05 were taken as the standard to define the DEGs.

### Survival analysis

Survival analysis was conducted on the union of DEGs, and a Kaplan–Meier diagram was drawn to illustrate the relationship between the overall survival of patients and the

expression level of the DEG genes. By using the log-rank test, the DEGs with $p < 0.05$ were defined as survival-related prognostic genes.

### PPI network

Survival-related prognostic genes were placed in the String database to retrieve the PPI network, identify the closely connected modules in the network, and define the closely connected module genes as the core prognostic genes.

### Enrichment analysis

Cluster Profiler was used to conduct functional enrichment analysis on relevant genes, and the Gene Ontology (GO) terms of significant enrichment were further identified according to the biological process (BP) functional enrichment analysis, while Kyoto Encyclopedia of Genes and Genomes (KEGG) pathway enrichment analysis was performed to identify the biological processes with significant enrichment. Enrichment analysis of KEGG pathways in GSEA was performed using the R package enrichplots for immune and matrix groupings ($P$ value cut-off $= 0.05$).

### Pivot analysis

The pivot node refers to (1) having at least two interactions with the module gene; (2) the significance analysis of the interaction between the node and each module should be less than or equal to 0.05, and the statistical method is hypergeometric.

Pivot analysis method of the ncRNA: The ncRNA-mRNA interaction relationship included in the RAID2.0 database is the background of the interaction. All of the interactions between ncRNAs and module genes were counted. Then, the interactions between each ncRNA and the in-module genes were counted.

Pivot analysis method of transcription factors (TFs): according to the interaction background of the human TF-mRNA regulation in the TRRUST v2 database, all TF interactions with the module genes were counted, then each TF interaction with the in-module genes and out-module genes were counted, and the pivot was screened according to the significance of the $p$ value of the hypergeometric test.

### Analysis of the hub genes

*Target gene-TF regulatory network.* Human TF-mRNA regulatory relationships were downloaded from the TRRUST V2 database to screen the 22 transcription factors that interacted with the hub genes.

*Target gene-miRNA regulatory network.* Human miRNA-mRNA targeting relationships were downloaded from the TargetScan (http://www.targetscan.org/vert_72/), miRTarBase database (https://mirtarbase.cuhk.edu.cn/~miRTarBase/miRTarBase_2022/php/index.php) and miRDB database (http://mirdb.org/). Based on the human miRNA-mRNA targeting relationships in the TargetScan, miRTarBase and miRDB databases, 22 hub gene targeting relationships were screened.

*Mutation analysis of the hub genes.* The R maftools package was used to analyze the single nucleotide variation (SNV) data of the hub gene samples obtained from TCGA.

The receiver operating characteristic (ROC) curve based on the hub genes: Based on the TCGA dataset, ROC curves were drawn for hub genes with a predict.time value of 1095.

*Immune infiltration analysis of the hub genes.* The proportion of immune cells in the colorectal cancer samples was estimated using the CIBERSORT and LM22 feature matrices. The Pearson correlation coefficient of the hub gene and the proportion of infiltration of the immune cells were calculated using the Psych and ar packages.

## RESULTS

### The matrix score was significantly correlated with stage T colorectal cancer

CRC expression profile data were downloaded from the TCGA database. Clinical information (OS>1 month) and immune and stromal scores calculated by the ESTIMATE algorithm were integrated. In addition, 362 CRC samples were finally retained for subsequent analysis. The distribution of the immune score and stromal score is shown in Fig. 1A. To determine the potential correlation between the overall survival rate and the immune and stromal scores, Maxstat software was used to find the optimal cut-offs for the immune and stromal scores. Cancer samples were divided into high and low scores according to the score of the optimal cut-off (stromal score $= -207.8886$/immune score $= -708.0627$), and the corresponding clinical information was used for survival analysis. In the survival curve of the immune score and matrix score, we found that the immune score and matrix score were significantly correlated with the survival time of the patients ($p = 0.0035$/ $p = 0.0034$, Figs. 1B–1C). At the same time, the results showed that the stromal score had no statistical significance with all stages, but it was significantly correlated with T staging in the correlation analysis ($p = 0.28$/ $p = 0.029$, Figs. 1D–1E). There was no statistical significance in the correlation between the immune score and staging. However, there was a certain difference in the scores for each stage ($p = 0.34$/ $p = 0.93$, Figs. 1F–1G).

### Comparison of colorectal cancer gene expression with immune score and stromal score

To reveal the correlations between gene expression and the immune and matrix scores, we evaluated the expression profile data of CRC patients in the TCGA database. A total of 228 DEGs were identified according to the matrix score (100 cases with a high score and 262 cases with a low score), among which 163 genes were upregulated and 65 genes were downregulated. The results showed that some of the DEGs were significantly different between the two groups (Fig. 2A). A total of 579 DEGs were identified according to the immune score groups (38 cases with high scores and 324 cases with low scores), among which 301 genes were upregulated and 278 genes were downregulated. These 789 DEGs were used as key genes for subsequent analysis (Fig. 2B), and their details are listed in Table S1. To clarify the potential function of the 789 DEGs, GO and KEGG enrichment analyses were performed on the DEGs. The results are shown in Figs. 3A and 3B (the top 8 were selected for display). GO analyses revealed that these DEGs were mostly involved in tissue-specific immune responses, positive regulation of adaptive immune

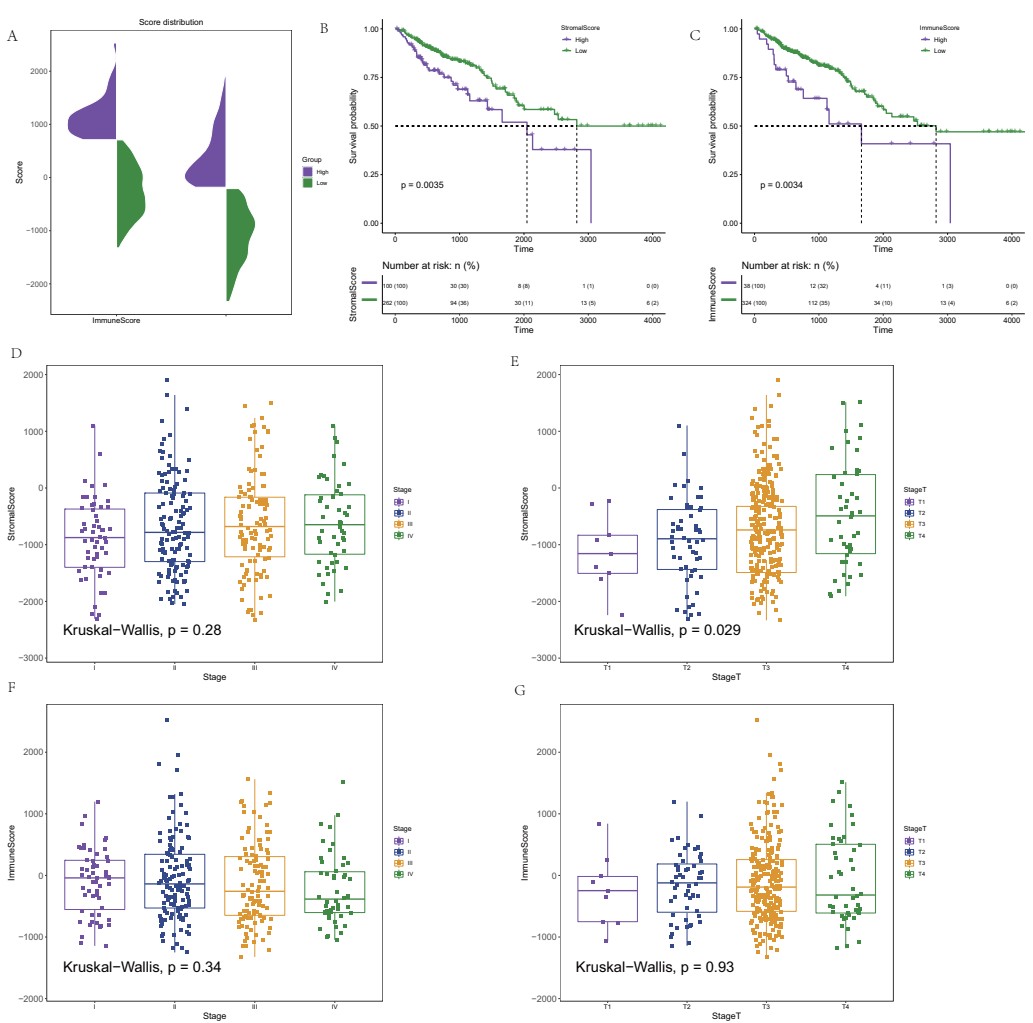

**Figure 1  The score distribution and the Kaplan–Meier survival curves and the correlation of immune score and stromal score and the stage of CRC.** (A) The distribution of immune score and stromal score. (B) Kaplan–Meier survival curve based on stromal score. (C) Kaplan–Meier survival curve based on immune score. (D) The correlation of stromals core and the stage of CRC. (E) The correlation of stromal score and the stage T of CRC. (F) The correlation of immune score and the stage of CRC. (G) The correlation of immune score and the stage T of CRC.

responses and so on. In addition, according to the KEGG pathway analyses, the DEGs were involved in the age-race signaling pathway in diabetic complications, cytokine-cytokine receptor interaction, IL17 signaling pathway and so on. The results of the enrichment analysis of KEGG pathways in GSEA showed that the genes were mostly enriched in the pathways of asthma, graft-versus-host disease, malaria, rheumatoid arthritis, and viral protein interaction with cytokines and cytokine receptors based on the ImmuneScore (Fig. 4A). In addition, the results showed that the genes were mostly enriched in the ECM-receptor interaction, hypertrophic cardiomyopathy, protein digestion and absorption,

A

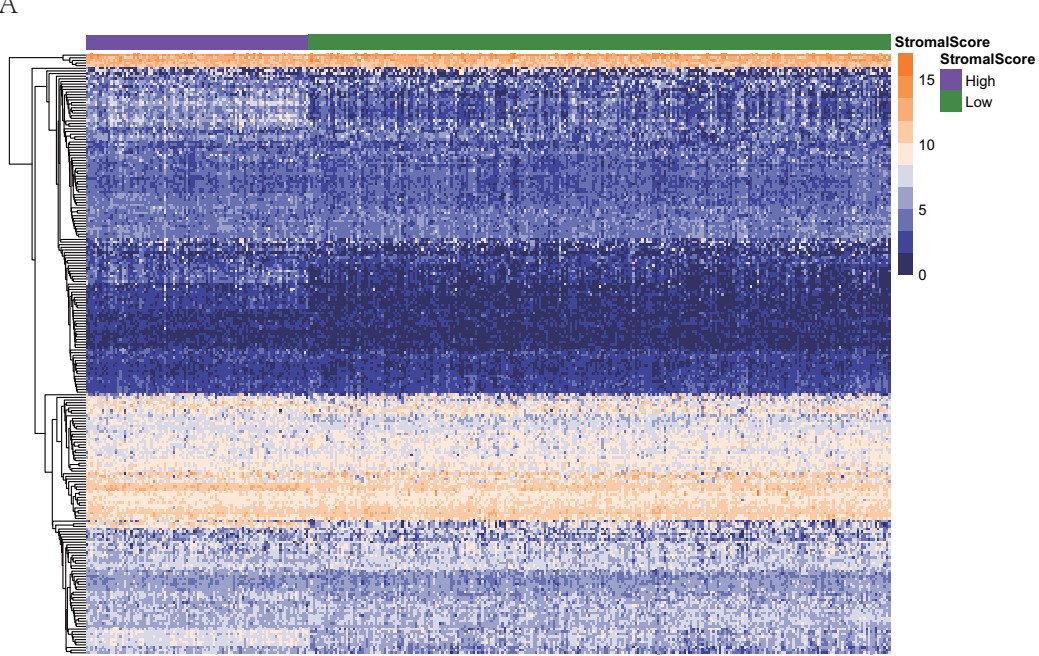

B

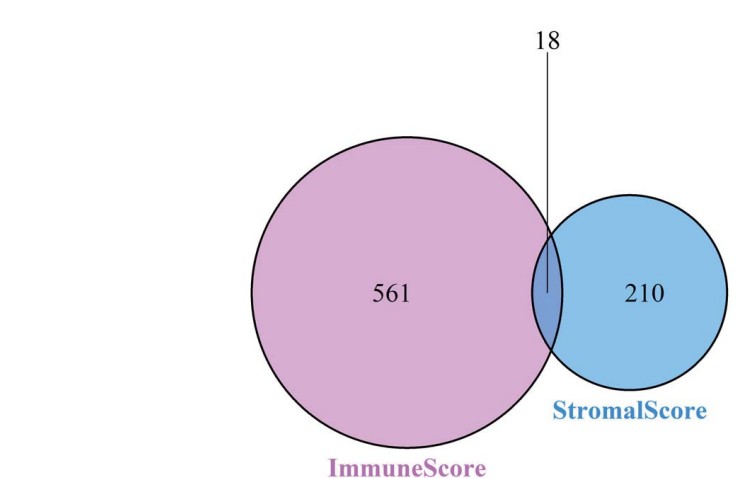

**Figure 2** **The heat map and venn diagram of ifferentially expressed genes.** (A) Heat map of differential gene expression in gene score. (B) Venn diagram. A total of 579 differentially expressed genes were identified according to the immune score groups (38 cases with high score and 324 cases with low score), among which 301 genes were up-regulated and 278 genes were down-regulated.

Staphylococcus aureus infection, and systemic lupus erythematosus pathways based on the Stromalscore (Fig. 4B).

## Survival analysis of DEGs

To screen out the genes associated with the prognosis of CRC, we divided the 789 DEGs into high and low expression groups according to their median expression and performed

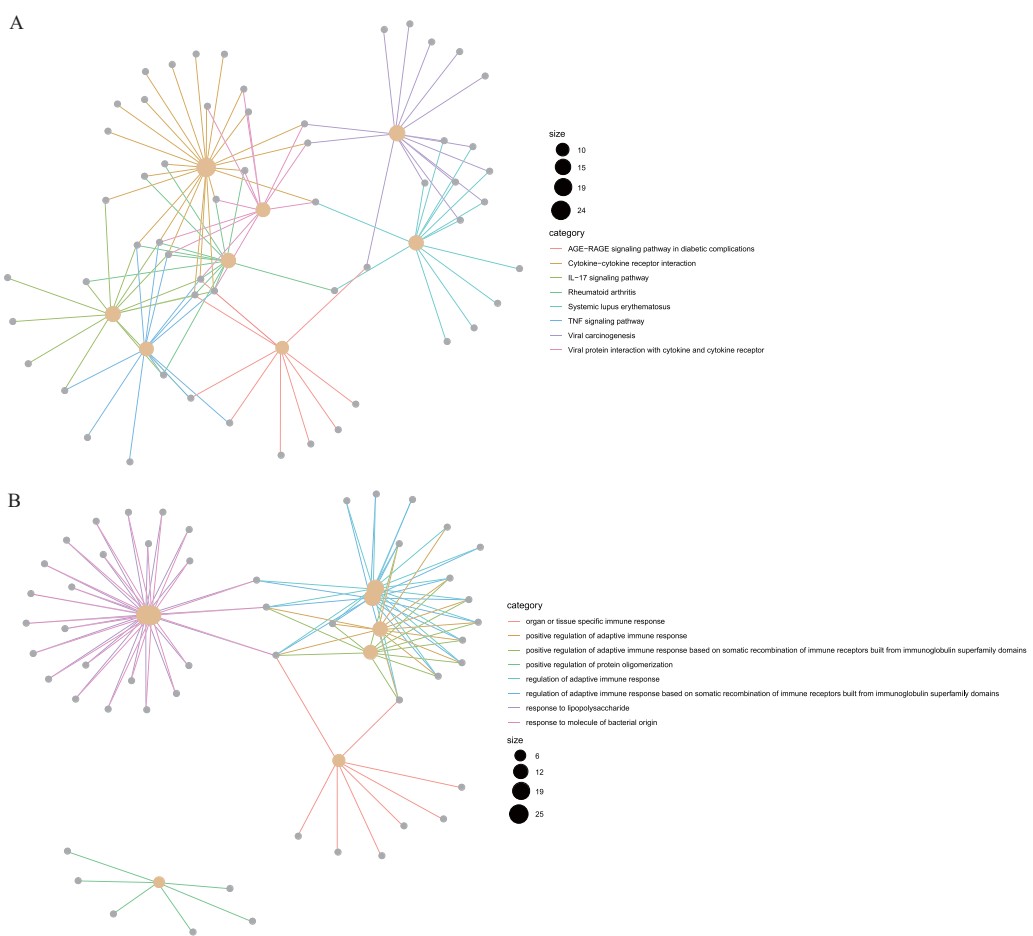

**Figure 3** **The KEGG and GO analysis of differentially expressed genes.** (A) KEGG analysis of the 789 differentially expressed genes (DEGs). (B) GO Biological Process of the 789 differentially expressed genes (DEGs).

survival analysis. A total of 121 survival-related prognostic genes were extracted ($p < 0.05$). The survival curves of some of the prognostic genes are shown in Fig. 5A.

## Construction of PPI networks between genes

To better understand the interactions between the identified prognostic genes, we used the STRING database to obtain a PPI network consisting of 35 nodes and 36 edges (Fig. 6). We identified tightly linked modules (green parts) in the network and defined the tightly coupled module genes as the core prognostic genes ($N = 22$). GO analysis and KEGG enrichment analysis were performed on 22 core prognostic genes. The results of the enrichment analysis are shown in Figs. 7A–7B (the top 10 genes were selected for display).

## Identification of the core prognostic genes

The 22 core prognostic genes obtained were validated in the GEO dataset, and all 22 genes were expressed normally. Survival analysis of the 22 core prognostic genes was performed

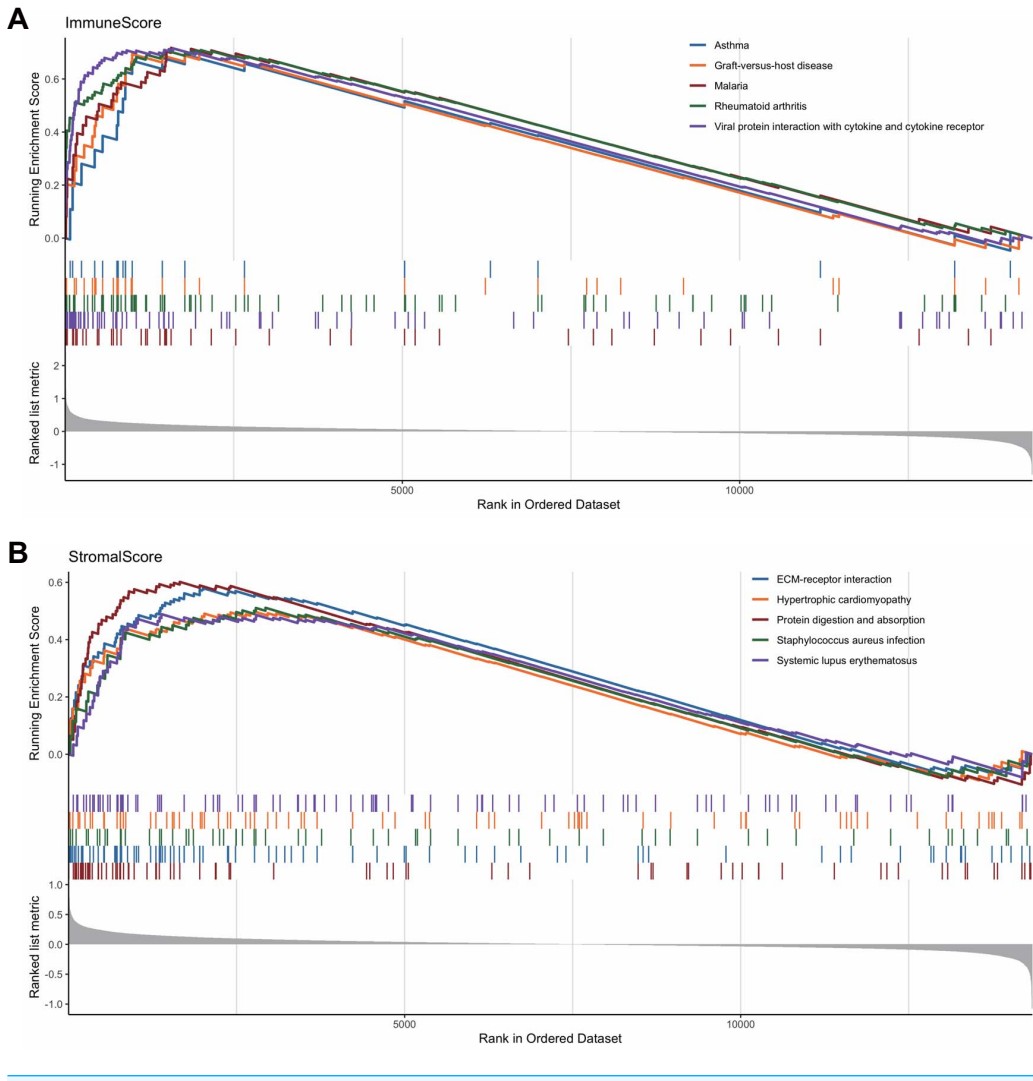

**Figure 4 The results of the enrichment analysis of KEGG pathway in GSEA.** (A) Enrichment analysis based on immunescore. (B) Enrichment analysis based on stromalscore.

in this dataset and identified only one gene consistent with the TCGA results ($P < 0.05$). The results of the gene survival analysis are shown in Fig. 5B.

## Regulation of the core prognostic genes by ncRNA/TF

Based on the 51913 ncRNA-mRNA interaction relationship in the RAID2.0 database, we searched for the pivot node (ncRNA) of the regulatory function module. When the $p$ value $< 0.05$, we obtained four ncRNAs that regulated the core prognosis gene (Table 3), including metastasis-associated lung adenocarcinoma transcript 1 (MALAT1) (associated with tumor cell proliferation and metastasis), colorectal neoplasia differentially expressed (CRNDE) (promotes colorectal cancer cell proliferation), FOXF1 adjacent noncoding developmental regulatory RNA (FENDRR) and taurine upregulated gene 1 (TUG1) (regulation of resistance to colorectal cancer methotrexate).

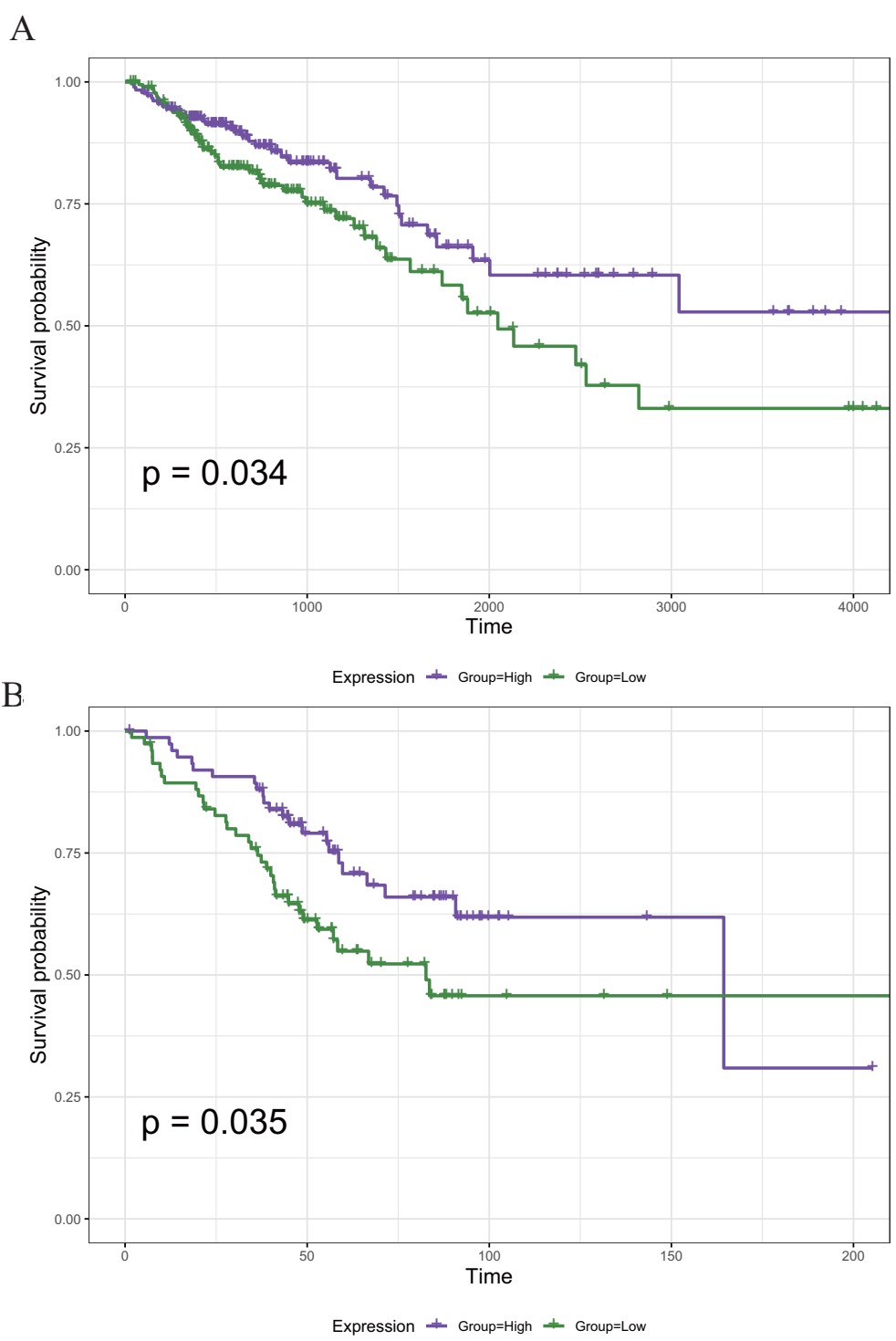

**Figure 5** **The Kaplan–Meier survival curve of the PRIM1 gene.** (A) Kaplan–Meier survival curve of the PRIM1 gene (TCGA database). A total of 121 survival-related prognostic genes were extracted ($p < 0.05$). (B) Kaplan–Meier survival curve of the PRIM1 gene (GEO database).

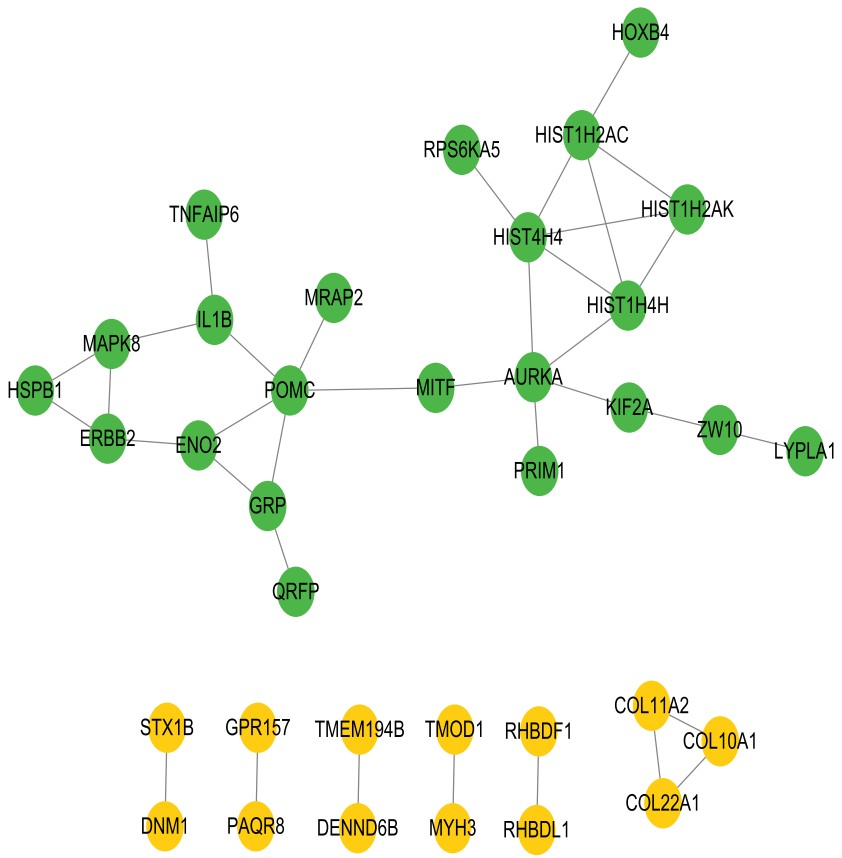

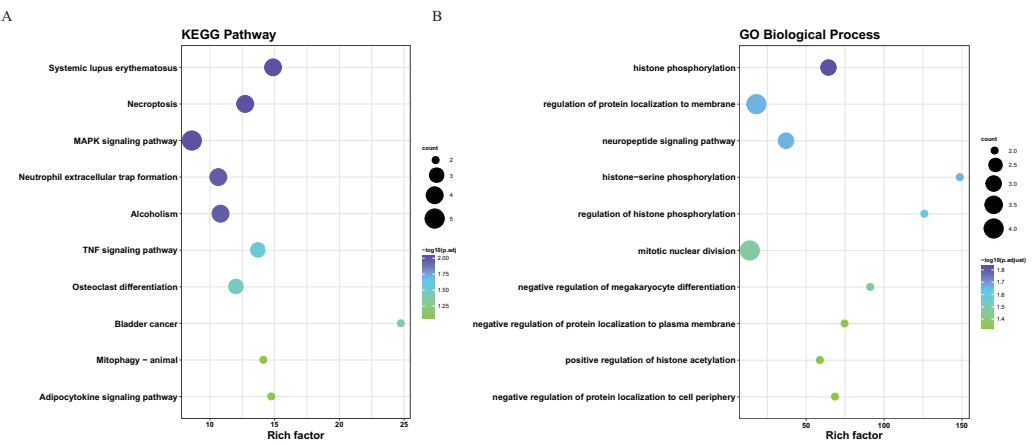

**Figure 6  PPI network.** The PPI network consisting of 35 nodes and 36 edges. A total of 22 core genes that associated with CRC prognosis were defined.

**Figure 7  The KEGG and GO analysis of 22 core prognostic genes.** (A) KEGG analysis of the 22 core prognostic genes. (B) GO Biological Process of the 22 core prognostic genes.

**Table 3  Pivot (ncRNA).**

| ncRNA | Connection | *P* value |
|-------|------------|-----------|
| MALAT1 | 7 | 0.000346134 |
| CANDE | 6 | 0.000878654 |
| FENDRR | 8 | 0.000451542 |
| TUG1 | 7 | 0.000286648 |

**Notes.**

Abbreviations: MALAT1, Metastasis-associated lung adenocarcinoma transcript 1; CRNDE, colorectal neoplasia differentially Expressed; FENDRR, FOXF1 adjacent non-coding developmental regulatory RNA; TUG1, taurine upregulated gene1.

**Table 4  Pivot (TF).**

| TF | Connection | *P* value |
|-----|------------|-----------|
| ATF1 | 3 | 0.000124081 |
| CEBPB | 4 | 0.000733944 |
| E2F3 | 2 | 0.00026685 |
| ETV4 | 2 | 0.001742111 |
| PAX3 | 3 | 6.68E−07 |
| SOX10 | 3 | 1.63E−05 |
| SP11 | 2 | 0.035007145 |
| DSF1 | 2 | 0.40192337 |
| USF2 | 2 | 0.18026951 |
| YBX1 | 2 | 0.004397422 |
| YY1 | 3 | 0.021654705 |

**Notes.**

Abbreviations: ATF1, activating transcription activator 1; CEBPB, CCAAT/enhancer binding proteins C/EBP beta; E2F3, E2F transcription factor 3; ETV4, ETS translocation variant 4; YBX1, Y-box protein 1; PAX3, Paired box 3; SOX10, SRY-related HMG-box 10; SP11, Synaptophysin 11; DSF1, Double-skin façade 1; USF2, Upstream stimulus factor 2; YY1, Yin Yang 1.

Based on the interaction relationships of 9396 human TF-mRNAs contained in the TRRUST v2 database, the pivot node (TF) of the regulatory function module was searched. When $p$ value < 0.05, 11 TFs with regulatory effects on the core prognostic genes were obtained (Table 4), including a number of cancer-related transcription factors, such as activating transcription activator 1 (ATF1), CCAAT/enhancer binding proteins C/EBP beta (CEBPB), E2F transcription Factor 3 (E2F3), ETS translocation variant 4 (ETV4), and Y-box protein 1 (YBX1). The visualization of the core prognostic genes and ncRNA/TF interactions is shown in Fig. 8A.

### Target gene-TF regulatory network of the 22 hub genes

Human TF-mRNA regulatory relationships were downloaded from the TRRUST V2 database to screen for transcription factors interacting with the 22 hub genes, including MYB, FOXD3, NFKB1, etc. (Fig. 8B).

### Target gene-miRNA regulatory network of the 22 hub genes

A total of 21,363,630 human miRNA-mRNA targeting relationships were downloaded from TargetScan. A total of 502,652 human miRNA-mRNA targeting relationships were downloaded from the miRTarBase database. A total of 1,102,737 human miRNA-mRNA

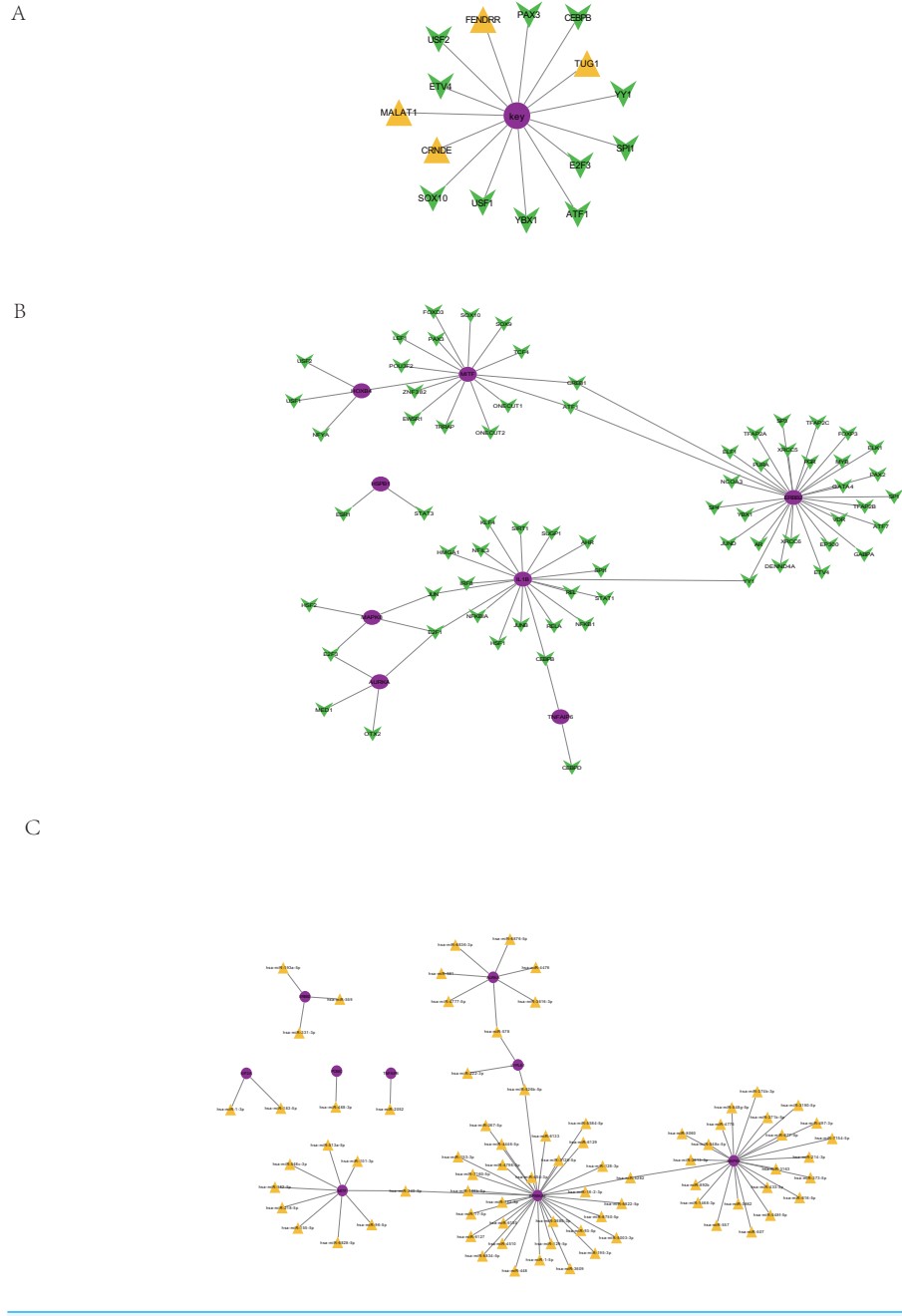

**Figure 8** **Regulation of core prognosis genes by ncRNA/TF, genes-TF and Target genes-miRNA regulatory network.** (A) Interaction between core prognostic genes and ncRNA/TF. (B) Target genes-TF regulatory network of the 22 hub-gene. Green for TF and red for Hubgene. (C) Target genes-miRNA regulatory network of the 22 hub-gene. Yellow represents mirna and purple represents Hubgene.

targeting relationships were downloaded from the miRDB database. Based on the human miRNA-mRNA targeting relationships in the TargetScan, miRTarBase and miRDB databases, 22 hub gene targeting relationships were screened, and 81 pairs of targeting relationships existed in all three databases (Fig. 8C).
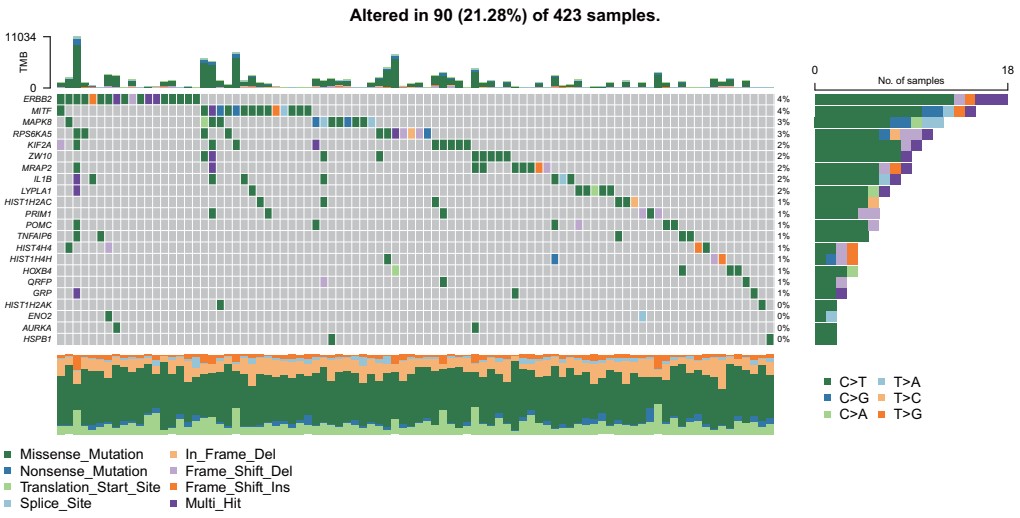

**Figure 9** Mutation analysis of hub genes.

## Mutation analysis of the hub genes

As shown in Fig. 9, the 22 hub gene samples had mutations only in 90 samples. Among these, ERBB2 had the highest mutation rate, most of which were missense mutations, and most of the samples had mutations of C > T.

## The ROC curve based on the hub genes

The area under the curve (AUC) of the 22 hub genes ranged from 0.344 to 0.669. Twelve hub genes, ENO2, GRP, HIST1H2AC, HIST1H2AK, HIST1H4H, HIST4H4, HOXB4, HSPB1, MIFT, POMC, QRFP, and TNFAIP6, had weak value in predicting the survival of colorectal cancer with >0.5 AUCs (Fig. 10).

## Immune infiltration analysis of the hub genes

As shown in Fig. 11, most of the hub genes were significantly correlated with plasma cells, M1 macrophages and other immune cells.

## Exploration of drugs for the treatment of colorectal cancer

For the core prognostic genes and related regulatory factors, we used the hypergeometric distribution test to screen related drugs in the context of drug-gene interactions in DrugBank. Finally, two markers of CRC regulation were obtained (Table 5). Among them, quercetin can significantly inhibit the action of cancer-promoting agents, inhibit the growth of isolated malignant cells, and inhibit the DNA, RNA and protein synthesis of Ehrlich ascites cancer cells.

## DISCUSSION

In the current work, we identified tumor microenvironment-related genes that contribute to CRC overall survival in the TCGA database and found that both the immune score and matrix score were significantly correlated with the patient survival time. Importantly,
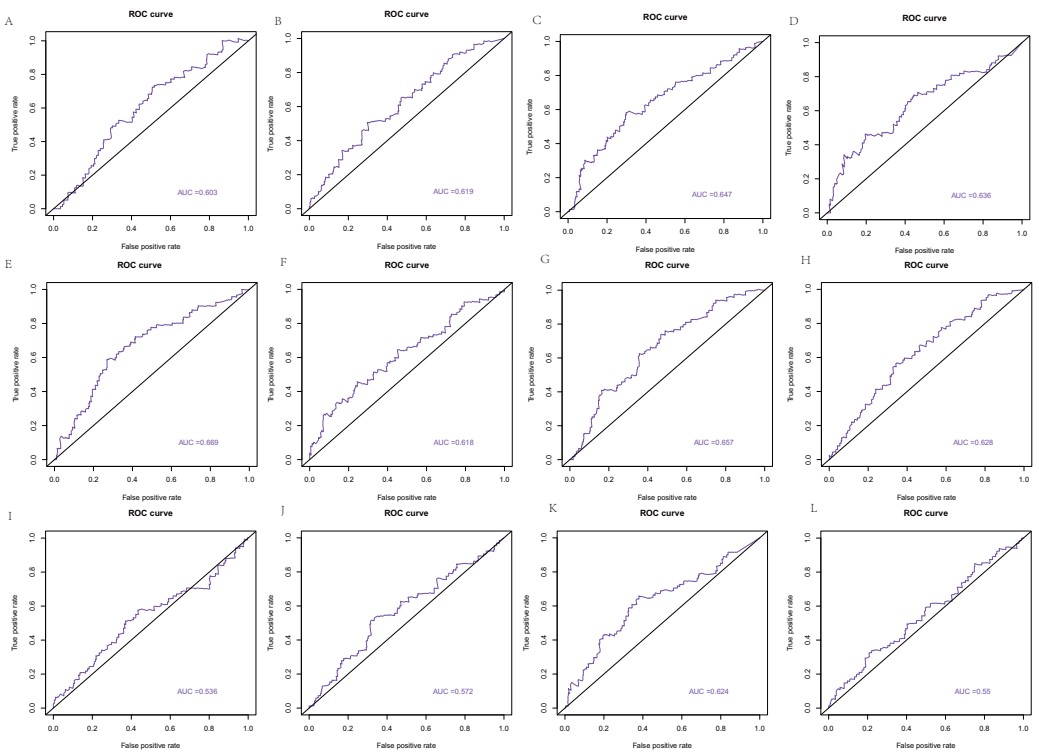

**Figure 10** **The ROC curve based on hub genes.** (A) ENO2, (B) GRP, (C) HIST1H2AC, (D) HIST1H2AK, (E) HIST1H4H, (F) HIST4H4, (G) HOXB4, (H) HSPB1, (I) MIFT, (J) POMC, (K) QRFP, (L) TNFAIP6.

we were able to validate 789 DEGs. In addition, 121 prognostic genes were found to be associated with survival. Twenty-two genes were identified as core prognostic genes. Four ncRNAs that regulate the core prognostic genes, 11 TFs that had a regulatory effect on the core prognostic genes, and 2 drugs that have regulatory effects on CRC were also identified.

Both the immune score and the matrix score can predict the purity of the tumor, as well as the number of stromal and immune cells. The more stromal and immune cells there are, the lower the purity of the tumor. In a study of immune scoring by Galon et al., patients with early TNM (stage I and II) were followed up for survival and recurrence of colon cancer, and it was found that patients with high immune scores had a longer survival period. Ninety-five percent of patients with a high score had no tumor recurrence within 18 years after surgery. Fifty percent of patients with low scores had a recurrence within 2 years after surgery (*Galon et al., 2012*). In 2018, Pages et al. reported that immune scores could provide a reliable assessment of the risk of recurrence of colon cancer, which supports the use of immune scores as part of the new TNM immunotyping (*Pagès et al., 2018*). Therefore, as a perfect supplement to the TNM staging system, the immune score will be a superior staging system for predicting the survival of tumor patients.

Searching for more metastatic genes related to CRC and studying their biological characteristics and the specific mechanism of metastasis can help control tumor metastasis,
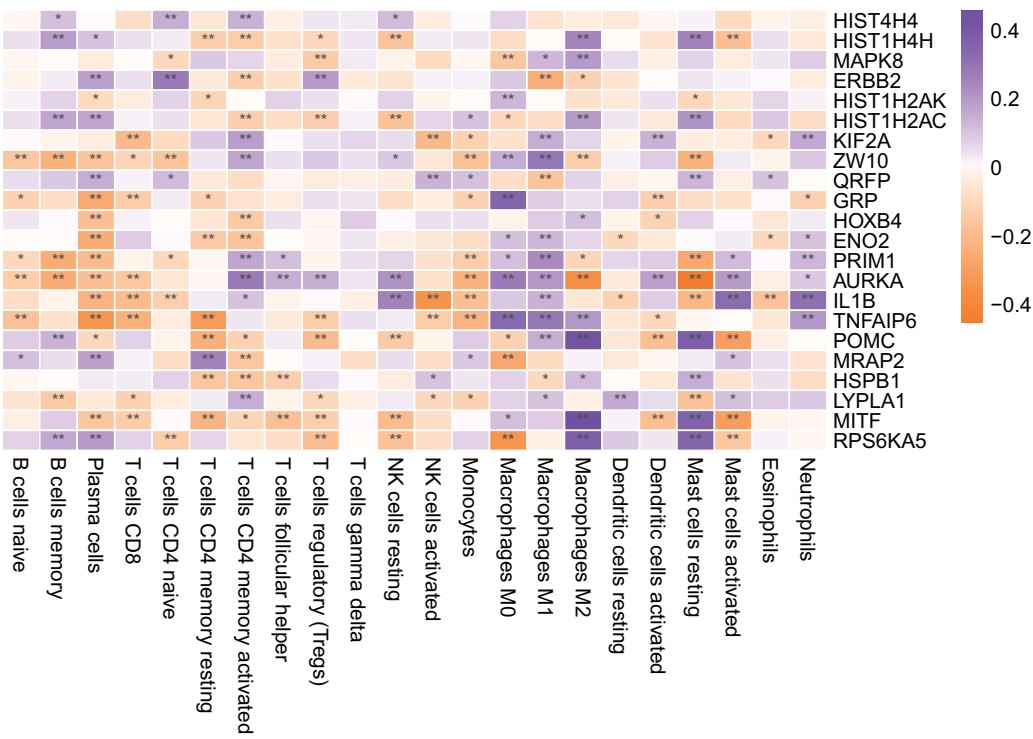

**Figure 11 Hub gene is associated with immune infiltrating cells.** An asterisk (*) means $0.01 \leq P < 0.05$, two asterisks (**) means $P < 0.01$.

**Table 5 Pivot (drug).**

| Drug | Connection | P value |
|---|---|---|
| Pseudoephedrine | 2 | 1.79E−05 |
| Quercetin | 2 | 9.57E−05 |

prevent recurrence and improve the prognosis of patients. In terms of multigene early diagnosis and a prognostic model of CRC, Marshall et al. obtained an early diagnostic model of CRC based on the gene expression profile of the peripheral blood of CRC patients (*Marshall et al., 2010*). *Sveen et al. (2012)* proposed a prognostic model by studying gene expression in tumor tissues, which has potential prognostic value in stage IV and V CRC patients. In addition, a large number of genes, including P53 (*Lüchtenborg et al., 2004*), K-ras (*Andreyev et al., 1990*), B-raf (*De Roock et al., 2010*), and N-ras (*Parsons et al., 2012*), have been shown to be associated with the prognosis of CRC.

In the present study, 22 genes were identified as the core prognostic genes of CRC. Among them, MAPK8 (*Slattery, Lundgreen & Wolff, 2012*), HSPB1 (*Nadin et al., 2012*), IL1B (*Sanabria-Salas et al., 2017*), PRIM1 (*Cloutier et al., 1997*) and so on were shown to be associated with the susceptibility to or prognosis of CRC. However, only the PRIM1 gene was validated in the GEO database.

PRIM1 (primase) plays a key role in the process of DNA synthesis initiation by synthesizing RNA primers for Okazaki fragments (*Cloutier et al., 1997*). Mutations in PRIM1 cause extensive apoptosis of retinal neurons through activation of the DNA damage checkpoint and tumor suppressor p53 (*Yamaguchi et al., 2008*). PRIM1 plays an important role in the strict control of the DNA replication fork during tumor cell proliferation and was shown to be involved in estrogen-induced breast cancer formation through activation of the G2/M cell cycle checkpoint (*Lee et al., 2018*). In addition, PRIM1 (*Job et al., 2018*) has been reported to be associated with CRC progression, and this gene is likely to be a prognostic marker of the CRC immune microenvironment. However, the mechanism by which PRIM1 affects CRC is not clear.

A total of four core ncRNAs were obtained that could regulate the core prognostic genes in the present study. MALAT1 is a long noncoding RNA with important biological functions and is located on chromosome llql3.1 (*Zhang et al., 2017*). MALT1 expression is upregulated in CRC tumor cells. A-kinase anchoring protein 9 (AKAP9) is a target gene for the cancer-promoting effect of MALAT1. AKAP9 is related to the polarization of cell spinning chains, and the overexpression of MALAT1 in normal intestinal epithelial cells causes them to divide and differentiate and promotes tumorigenesis (*Xu et al., 2011*). *In vitro* experiments have demonstrated that its knockout can inhibit the migration of p-catenin from the cytoplasm to the nucleus, resulting in decreased expression of c-MYC and MMP-7, whereas overexpression of MALAT1 can activate the Wnt/p-catenin pathway and promote the invasion and metastasis of CRC (*Ji et al., 2013*). *Zheng et al. (2014)* further confirmed that MALAT1 was significantly upregulated in CRC tissues and was associated with a poor prognosis in stage II/III CRC patients.

CRNDE was first identified as a biomarker of CRC. High expression of lncRNA CRNDE was detected in more than 90% of colorectal adenomas and adenocarcinoma cells (*Ning et al., 2014*). Experiments showed that the regulatory effect of CRNDE was related to the IGF signaling pathway and was involved in the carcinogenic process of intestinal epithelial cells (*Huan et al., 2017*). CRNDE can activate the Wnt/x-catenin signaling pathway in CRC patients, thus promoting tumor cell proliferation, invasion and other processes (*Clevers & Nusse, 2012*). In addition, *Li et al. (2018)* showed that CRNDE can be used as a potential noninvasive serum marker to predict the efficacy of first-line FOLFOX regimens for metastatic colorectal cancer and it is associated with a poor prognosis. Although CRNDE can enhance the proliferation, migration and invasion of CRC cell lines, the mechanism by which CRNDE promotes the proliferation and migration of CRC cells remains unclear.

Taurine upregulated gene 1 (TUGl) is a newly discovered carcinogenic lncRNA located on chromosome 22q12 (*Li et al., 2016*). Numerous studies have demonstrated that TUGl expression levels are significantly elevated in CRC tissues and cell lines. The expression level of TUGl in CRC tissues was 4-6 times higher than that in adjacent noncancerous tissues (*Sun et al., 2016*). *In vitro* tests, overexpression of TUGl can induce the formation of tumor cell colonies and activate the expression of EMT-related genes to improve the invasion and metastasis of tumors and promote liver metastasis of CRC (*Sun et al., 2018*). TUGl promotes cancer cell proliferation, migration, invasion, and epithelial-mesenchymal transition (EMT) and inhibits tumor cell apoptosis through complex mechanisms,

including competitive inhibition of miRNA function. Nevertheless, the specific oncogenic mechanism of TUGl remains to be further elucidated (*Huang et al., 2016*).

Transcription factors activate the transcription of downstream target genes by binding to the target gene promoter region. Studies have found that transcription factors are closely related to tumors. A total of 11 TF genes regulating the core prognosis were obtained in the present study. We are particularly interested in ATF1, CEBPB, E2F3, and ETV4. ATF1 belongs to the cAMP response element binding protein (cyclic AMP response element binding protein, after CREB) family (*Huang et al., 2016*). A study found that ATF1 in a wide variety of tumors plays a role as both an oncogene and a tumor suppressor gene by influencing the cell signal transduction pathways involved in tumor cells related to biological processes, such as proliferation, apoptosis, angiogenesis, migration, invasion, and immune surveillance, which affects the occurrence and development of tumors (*Pu, Storr & Ahmad, 2018*). In mouse colorectal cancer cells, active phosphorylated CREB is elevated (*Sampurno et al., 2013*), and p300 can assist the phosphorylation of CREB and activate intestinal stem cell transcription factors, including Myb, that regulate the proliferation of intestinal epithelial cells (*Ramaswamy et al., 2018*). It was speculated that ATF1 is involved in human colorectal cancer cell proliferation through the p300-MYB-CREB axis.

The transcription factor CEBPB is involved in a number of biological processes, including cell differentiation, metabolic balance, proliferation, tumorigenesis, apoptosis, immune and stress responses, energy metabolism, and blood production (*Ramji & Foka, 2002*). Some studies have shown that CEBPB affects tumorigenesis by interacting with other genes to form a regulatory network during tumorigenesis and development (*Abreu & Sealy, 2010*). Other studies have also shown that the LIP subtype of the C/EBP transcription factor can induce apoptosis of human breast cancer cells and induce its own phagocytosis, which may be acting as a tumor suppressor by inducing tumor autophagy (*Koslowski et al., 2009*).

Additionally, the E2F transcription factor is an important regulator of G1 phase entry into S phase in the cell cycle, and it is closely related to tumor occurrence and cell apoptosis (*Leone et al., 1998*). *Hurst et al. (2008)* showed that E2F3 inhibited pRb and P53 through two important pathways of cell proliferation regulation and tumor monitoring, p16(Ink4a)-cycd/cdk4-rb-e2f and Arf/mdm2-p53, respectively, leading to the disorder of cell cycle regulation and thus promoting the occurrence and development of tumors. *Wang, Zhao & Yuan (2011)* found that the expression of E2F3 was high in colorectal carcinoma. There is a positive correlation with the expression of E2F3 in colorectal carcinoma.

Furthermore, ETV4 is a member of the polyomavirus enhancer activator 3 (PEA3) subfamily of ETS transcription factors, in which transcription factors can recognize and bind to GGAA/T sequences to regulate the expression of multiple target genes, thus affecting the occurrence and development of diseases. Existing studies have found that ETV4's ability to promote cancer migration is closely related to MMP (*Fung et al., 2016*). For example, in breast cancer, ETV4 can promote the migration of cancer cells by promoting the expression of MMP2 (*Bièche et al., 2004*); in esophageal cancer, ETV4 can promote the metastasis of cancer cells by promoting the expression of MMP1 (*Keld et al., 2010*). In addition, ETV4 was found to promote the invasion and metastasis of cancer cells by promoting the greening

of COX2 (*Subbaramaiah, 2002*). Of course, the mechanisms of the pro-proliferative and pro-migratory effects of ETV4 in CRC are not fully understood.

Notably, we also predicted that quercetin might be a drug that regulates CRC. Quercetin is a flavonoid with a wide range of biological activities, including antioxidant activity, scavenging oxygen free radicals, antifibrosis activity, lowering blood pressure, lowering blood glucose, protecting heart muscle and antitumor activity (*Nguyen et al., 2017*). In recent years, many studies have found that quercetin can inhibit a variety of cancer cells, especially colon cancer. Quercetin can not only inhibit the proliferation and induce the apoptosis of colon cancer cells but also reduce the number of abnormal gland crypts in the colon (*Miyamoto, Yasui & Ohigashi, 2010*). However, there are few studies on the effect of quercetin on colon cancer, and the specific molecular mechanism is still not fully understood. *Luo et al. (2014)* reported that quercetin may inhibit cell proliferation and induce apoptosis through the bcl-2 and c-myc genes. *Park et al. (2005)* found that quercetin could significantly inhibit the transcriptional activity of catenin/Tcf signals in SW480 colon cancer cells by reducing the protein levels of large-catenin and Tcf in the nucleus. Thus, quercetin can be used as an adjuvant drug to inhibit the growth of colon cancer cells and has potential value in the drug treatment of colon cancer.

## CONCLUSIONS

In summary, from the functional enrichment analysis of the TCGA database applied by ESTIMATE algorithm-based immune scores, we extracted a list of tumor microenvironment-related genes. These genes could be useful for determining the prognosis of CRC patients. In addition, further investigation of these genes as well as their regulators, including ncRNAs and TFs, provides a stronger predictor of survival than individual genes. Finally, the detection of drugs that regulate CRC could lead to novel insights into potential treatments for CRC.

### Funding
This work was supported by the Basic Public Welfare Research Program of Zhejiang Province, China (LGF18H160033), the Medical and Health Science and Technology Project of Zhejiang Province, China (2019KY214), the Medical and Health Science and Technology Project of Zhejiang Province, China (2019KY692), Project of Public Welfare research of Jiaxing (2019AD32257), and the Jiaxing Key Discipline of Medicine-Oncology (Supporting Subject) (2019-zc-11). The funders had no role in study design, data collection and analysis, decision to publish, or preparation of the manuscript.

### Grant Disclosures
The following grant information was disclosed by the authors:
Basic Public Welfare Research Program of Zhejiang Province, China: LGF18H160033.
Medical and Health Science and Technology Project of Zhejiang Province, China: 2019KY214.

Medical and Health Science and Technology Project of Zhejiang Province, China: 2019KY692.

Project of Public Welfare research of Jiaxing: 2019AD32257.

Jiaxing Key Discipline of Medicine-Oncology (Supporting Subject): 2019-zc-11.

## Competing Interests

The authors declare there are no competing interests.

## Author Contributions

- Yi Zhu conceived and designed the experiments, prepared figures and/or tables, authored or reviewed drafts of the paper, and approved the final draft.
- Yuan Zhou, HongGang Jiang and ZhiHeng Chen performed the experiments, analyzed the data, authored or reviewed drafts of the paper, and approved the final draft.
- BoHao Lu performed the experiments, authored or reviewed drafts of the paper, and approved the final draft.

## Data Availability

Raw data are available in the Supplementary Files.

## Supplemental Information

Supplemental information for this article can be found online at http://dx.doi.org/10.7717/peerj.12452#supplemental-information.

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
