# Peer review of "Analysis of core genes for colorectal cancer prognosis based on immune and stromal scores"

_PeerJ, doi:10.7717/peerj.12452_

## Round 0.1 · original submission · Major Revisions

Please make corrections and resubmit the manuscript.

Reviewer 1 ·

Basic reporting

The title, introduction, methods, results and discussion are appropriate for the content of the text. Furthermore, the article is well constructed, the experiments are well conducted, and analysis is well performed. The figures are relevant, high quality, well labelled and described. The abstract needs more editing with more details and summary level sentences.

Experimental design

The experimental design is original and the research is within the scope of the journal. Research question is well defined, relevant and meaningful. The methods are highly technical, ethical and logistical. Statistical methods are chosen correctly.

Validity of the findings

All underlying data have been provided in detail. The findings are meaningful. The conclusions are well stated and relevant to the research questions.

Additional comments

This paper investigates the function of expression of tumor micro-environment-related genes in overall survival(OS) and prognosis of colorectal cancer(CRC) by analyzing TCGA datasets. The authors identified avlist of tumor micro-environment-related genes that strongly correlate with the prognosis of CRC patients by a series of bioinformatic analysis. Moreover, the authors validate their findings in a GEO dataset. Furthermore, the authors explore the related pathways through enrichment analysis. In short, this study identified several tumor micro-environment-related genes as potential predictors for the prognosis of CRC.

Editorial Criteria
BASIC REPORTING
The title, introduction, methods, results and discussion are appropriate for the content of the text. Furthermore, the article is well constructed, the experiments are well conducted, and analysis is well performed. The figures are relevant, high quality, well labelled and described. The abstract needs more editing with more details and summary level sentences.
EXPERIMENTAL DESIGN
The experimental design is original and the research is within the scope of the journal. Research question is well defined, relevant and meaningful. The methods are highly technical, ethical and logistical. Statistical methods are chosen correctly.
VALIDITY OF THE FINDINGS
All underlying data have been provided in detail. The findings are meaningful. The conclusions are well stated and relevant to the research questions.

Overall, I think this paper is novel and will be of interest to the community of lung cancer genetics, especially CRC research. The statistical part is valid and makes sense. The authors make it comprehensive by integrating analysis of multiple sources including GEO and TCGA. The main strengths of this paper is that it addresses an interesting and unexplored question, finds a novel discovery based on a carefully selected set of bioinformatic procedures. As such this article represents an excellent and elegant bioinformatics study which will almost certainly influence our thinking about the function of ferroptosis-related genes in CRC. Some of the weaknesses are the lack of in vitro or in vivo validation experiments. In general, the work is convincing except some major and minor comments below:


Major Comments:

The Methods of the Abstract are too simple, I would recommend expanding it and adding details like what datasets were used, what the aims are for PPI, GO, KEGG, Pivot analysis, etc, what the discovery dataset is, what the validation dataset is.

The Conclusions of the Abstract are too simple. Please add one more sentence about the importance of this study.

Line 85 to 90: Please add sample size and sequencing platforms for both TCGA and GEO datasets.

I do see multiple normalization methods for COAD (Colon adenocarcinoma) samples in GDC, which includes HTSeq - Counts (sample size: 514), HTSeq - FPKM(sample size: 514) and HTSeq - FPKM-UQ(sample size: 514). Please explain why the sample size in the manuscript is 362 rather than 514. Please also explain which normalization method was chosen for this analysis and reason.
(https://portal.gdc.cancer.gov/repository?facetTab=files&filters=%7B%22op%22%3A%22and%22%2C%22content%22%3A%5B%7B%22op%22%3A%22in%22%2C%22content%22%3A%7B%22field%22%3A%22cases.primary_site%22%2C%22value%22%3A%5B%22colon%22%5D%7D%7D%2C%7B%22op%22%3A%22in%22%2C%22content%22%3A%7B%22field%22%3A%22cases.project.program.name%22%2C%22value%22%3A%5B%22TCGA%22%5D%7D%7D%2C%7B%22op%22%3A%22in%22%2C%22content%22%3A%7B%22field%22%3A%22cases.project.project_id%22%2C%22value%22%3A%5B%22TCGA-COAD%22%5D%7D%7D%2C%7B%22op%22%3A%22in%22%2C%22content%22%3A%7B%22field%22%3A%22files.data_category%22%2C%22value%22%3A%5B%22transcriptome%20profiling%22%5D%7D%7D%2C%7B%22op%22%3A%22in%22%2C%22content%22%3A%7B%22field%22%3A%22files.experimental_strategy%22%2C%22value%22%3A%5B%22RNA-Seq%22%5D%7D%7D%5D%7D)


I’m wondering if there are any ongoing clinical trials focusing on micro-environment associated genes identified in this study in CRC? It will be very strong evidence for the significance of the current study if so.



Minor Comments:
Line 32: please add “(CRC)” after “Colorectal cancer”.

Line 38, 39: please add the full name of GO and TF.

Please make sure all the gene names should be italic.

Line 86 to 87: please replace “Gene expression profile data and clinical information of colorectal cancer patients were obtained from TCGA database” with “Gene expression profile data and clinical information of colorectal cancer patients from TCGA was obtained from NCI Genomic Data Commons (https://portal.gdc.cancer.gov)”. Since GDC is the official data host for TCGA data. Please also add this paper as reference for GDC: https://www.nature.com/articles/s41588-021-00791-5 .

Line 166 to 167: I would recommend adding more details about the top hits for the results of the enrichment analysis.

It would be great if there is a session of abbreviations to list all the abbreviations for the database names. I would also recommend including abbreviations like CRC, TCGA, GEO, DEG, GO, KEGG etc in that list.

Annotated reviews are not available for download in order to protect the identity of reviewers who chose to remain anonymous.

Reviewer 2 ·

Basic reporting

Missing words in sentences at various paragraphs. For example: "In 2011, scientist began to various stages (TNMI - IV) of patients with". Please revise the manuscript to check for consistency in grammar.

Experimental design

The tools and software used in the manuscript is in accordance with the aim of experiment.

Validity of the findings

Please see the below comments

1) Please try GSEA on the expressed genes. It may further output interesting pathways and gene sets
2) How many genes were used for GO analysis?
A small number of genes (22 prognostic genes) used for the GO analysis are too few. Please explain the results.

Additional comments

Overall the authors have done a good work with experimental design and providing evidence. Please revise the manuscript with recommendation as stated in the previous sections.

Reviewer 3 ·

Basic reporting

1.Poor English expression and lack of technical terminology
2.Many relevant articles not cited

Experimental design

Lack of innovation in experimental design, unconvincing results and lack of validation

Validity of the findings

Lack of innovation in experimental design, unconvincing results and lack of validation

Additional comments

The authors identified prognostic genes in colorectal cancer. There are several important flaws in the article that I believe cannot be published in PeerJ
1. This paper is a purely bioinformatic analysis, and in recent years there have been so many similar studies that this paper cannot provide additional value.
2. The title of the article is significantly flawed. How do we know that these genes are genes of the tumor microenvironment? Tumour cells do not express these genes?
3. A single algorithm is used in this paper, and it is recommended that 2 similar algorithms be added to verify the conclusions of this paper.
4. The lack of experimental validation in this paper is an important flaw.
5. In this paper, GSE102479 is used as a validation. It is recommended that all colorectal cancer-related data from the GEO database be included for validation, and to my knowledge, there should be >10 similar datasets.
6. The pictures in this article are not aesthetically pleasing and reflect a lack of technical skills as a bioinformatics analyst.

---

## Round 0.2 · Major Revisions

The manuscript needs some more clarification as stated above.

Reviewer 1 ·

Basic reporting

The languages still need to be polished a little bit.

Experimental design

The experimental design is original and the research is within the scope of the journal. Research question is well defined, relevant and meaningful. The methods are highly technical, ethical and logistical. Statistical methods are chosen correctly.

Validity of the findings

All underlying data have been provided in detail. The findings are meaningful. The conclusions are well stated and relevant to the research questions.

Additional comments

The revised version of the manuscript has addressed all the comments I raised.

Annotated reviews are not available for download in order to protect the identity of reviewers who chose to remain anonymous.

Reviewer 2 ·

Basic reporting

Professional English used.

Experimental design

The bioinformatics tools applied are in accordance with the design of experiment.

Validity of the findings

Some parts of analysis still needs clarification. For example:
1) The statement "The 22 core prognosis genes obtained were validated in the GEO data set, and 22 genes were expressed.
183 Further analysis of survival in this dataset identified only one gene consistent with TCGA results (P<0.05).
184 The results of gene survival analysis are shown in Figure 4B. "

How 22 core prognosis genes were validated in GEO dataset. Did you see the expression of these genes? If yes, please upload the expression/violin plots

2) For GSEA, you would want to perform the analysis on whole gene list as GSEA ranks the gene list and performs the pathway enrichment analysis.

Additional comments

The manuscript needs some more clarification as stated above.

---

## Round 0.3 · Minor Revisions

Please make these minor corrections and we will soon approve your manuscript.

Reviewer 1 ·

Basic reporting

The title, introduction, methods, results and discussion are appropriate for the content of the text. Furthermore, the article is well constructed, the experiments are well conducted, and analysis is well performed. The figures are relevant, high quality, well labelled and described. The abstract needs more editing with more details and summary level sentences.

The authors have carefully revised the languages by a English native speaker. All the revisions were reviewed and look fine to me.

Experimental design

The experimental design is original and the research is within the scope of the journal. Research question is well defined, relevant and meaningful. The methods are highly technical, ethical and logistical. Statistical methods are chosen correctly.

Validity of the findings

All underlying data have been provided in detail. The findings are meaningful. The conclusions are well stated and relevant to the research questions.

Additional comments

The revised version of the manuscript has addressed all the comments I raised. The updated manuscript looks fine to me.

Reviewer 2 ·

Basic reporting

Grammatical mistakes in the sentence formation
Ex , for line number 170
The results of the enrichment analysis of KEGG pathway in GSEA shown that the gene ..

Similarly for line number 172
In addition, the results shown that the …

Please correct these sentence formation. Please also thoroughly check the manuscript for grammatical errors.

Experimental design

Methods have been described.

Validity of the findings

Conclusion are stated. Limited to supporting results

Additional comments

This manuscript informs the scientific community about prognostic genes in CRC using various statistical and bioinformatics approach.

---

## Round 0.4 · accepted · Accept

Congratulations on the manuscript that is now accepted and will soon be published.